# Long Range Language Modeling via Gated State Spaces

**Harsh Mehta**
Google Research
harshm@google.com

**Ankit Gupta**
IBM Research[*]
ankitgupta.iitkanpur@gmail.com

**Ashok Cutkosky**
Boston University
ashok@cutkosky.com

**Behnam Neyshabur**
Deepmind
neyshabur@google.com

## Abstract

State space models have shown to be effective at modeling long range dependencies, especially on sequence classification tasks. In this work we focus on autoregressive sequence modeling over English books, Github source code and ArXiv mathematics articles. Based on recent developments around the effectiveness of gated activation functions, we propose a new layer named *Gated State Space* (GSS) and show that it trains significantly faster than the diagonal version of S4 (i.e. DSS) on TPUs, is competitive with several well-tuned Transformer-based baselines and exhibits zero-shot generalization to longer inputs while being straightforward to implement. Finally, we show that leveraging self-attention to model local dependencies improves the performance of GSS even further.

## 1 Introduction

Modeling long range dependencies on sequential data is a crucial step towards closing the gap with human-level performance on many tasks. Attention based models like Transformer (Vaswani et al., 2017) have proven to be a strong choice of backbone architecture for a considerable number of tasks across modalities and scale (Devlin et al., 2019; Brown et al., 2020; Dosovitskiy et al., 2021). Vanilla Multi-Head-Attention famously incurs $\Omega(L^2)$ penalty in modeling a sequence of length $L$. This is prohibitive at best for tasks where the model is required to capture long range dependencies from various parts of the input. Over the years, a variety of improvements have been proposed to alleviate this quadratic complexity (Tay et al., 2020; Choromanski et al., 2021; Ramsauer et al., 2021; Wang et al., 2020; Katharopoulos et al., 2020; Peng et al., 2021; Ainslie et al., 2020; Zaheer et al., 2020; Beltagy et al., 2020; Dai et al., 2020; Kitaev et al., 2020; Vyas et al., 2020).

On a somewhat orthogonal direction, attention-free models based on state spaces, such as S4 (Gu et al., 2022a) and DSS (Gupta et al., 2022), have shown remarkable improvements on Long Range Arena (LRA) (Tay et al., 2021), a benchmark designed with long range modeling as its focus and consists of diverse tasks with 1k-16k sequence length across modalities. These models require careful initialization, originally borrowing ideas from the theory of HiPPO matrices (Voelker et al., 2019; Gu et al., 2020), to achieve good results on LRA.

In this work, we explore and extend the use of state space models by focusing solely on the task of autoregressive sequence modeling (Brown et al., 2020; Rae et al., 2021; Chowdhery et al., 2022; Zhang et al., 2022; Hoffmann et al., 2022; Srivastava et al., 2022). Several key properties endowed by the state space model family makes it particularly attractive in the context of language modeling. First, it reduces the $\Omega(L^2)$ complexity on input sequence length to $O(L \log L)$. This complexity results from the use of Fast Fourier Transform (FFT) (Cooley & Tukey, 1965) for performing convolutions, which will be described in detail in later sections. Second, the state space model is fully parallelizable in the length dimension. This is an arguably subtle but an important property at training time. Note

---

[*]Work done at Tel-Aviv University

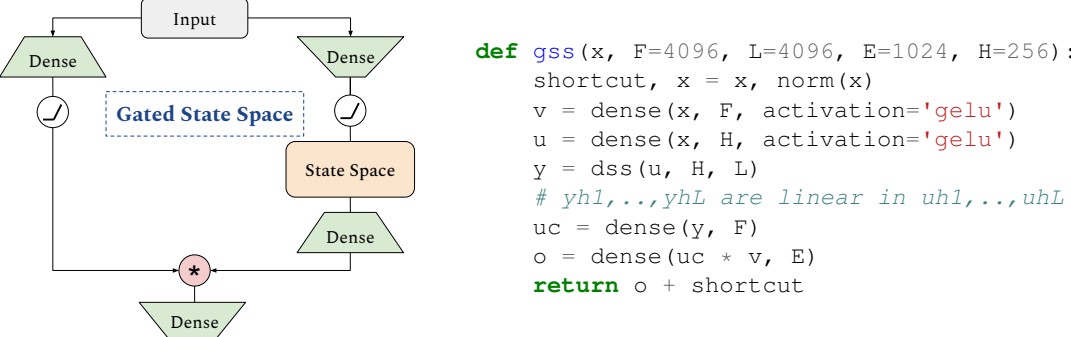

```python
def gss(x, F=4096, L=4096, E=1024, H=256):
    shortcut, x = x, norm(x)
    v = dense(x, F, activation='gelu')
    u = dense(x, H, activation='gelu')
    y = dss(u, H, L)
    # yh1,..,yhL are linear in uh1,..,uhL
    uc = dense(y, F)
    o = dense(uc * v, E)
    return o + shortcut
```

Figure 1: (a) Our proposed Gated State Space (GSS) layer, (b) Pseudocode for GSS (full implementation in §A.2).

that transformers are also fully parallelizable, a worthy advantage over traditional RNNs for modeling sequences, which otherwise incurs only an $O(L)$ penalty. While this parallelism is useful at training time, it may also be a curse at inference time where decoding every token requires attending to the whole past. The ideal model is parallelizable at training time but incurs a small constant cost (per decoded token) at inference time. This brings us to the final point. Due to the inherent convolution-recurrence equivalence of the state space model, it can be made to accumulate state and unroll like an RNN at inference time without any approximations.

Despite these attractive properties, we found that current state space models (e.g. S4, DSS) run slower than we expected on TPUs, our accelerator of choice. We take this opportunity to modify the architecture to reduce dimensionality of specific bottleneck operations. Our proposed changes borrow from a well-supported empirical success of gating units (Shazeer, 2020). Specifically, Hua et al. (2022) observed that replacing the Feed-Forward layer in the Transformer with gating units allows for a reduced dimensionality when mixing tokens along the length dimension using self-attention. We extend the use of gating units to state space model family and observe that, even in our context, the use of gating units allows for a reduction in dimensionality when performing FFT operations, which we observed to be the main bottleneck behind slow training. Furthermore, somewhat contrary to observations made by S4 and DSS authors, we found the performance on language modeling tasks to be much less sensitive to initialization: only the scale and structural aspects of initialization of state space variables were important and not the exact values. We were able to successfully train the model while initializing the state space variables randomly. This departs from the reliance of the design on the theory of HiPPO matrices, which led the S4 model to employ several numerical linear algebra tricks to able to make it work. Combining both of these contributions, we propose a layer named Gated State Space (GSS) (Figure 1), which we empirically verified to be 2-3× faster than DSS while keeping the perplexity on several language modeling benchmarks (Table 1).

Going one step further, we also perform a comparison with well-tuned and performant baselines reported in Block Recurrent Transformers (Hutchins et al., 2022), on several long range language modeling benchmarks over modalities such as English books, raw source code from Github and LaTeX source of ArXiv mathematics articles. As detailed in Table 2, while our GSS model currently lags behind on some tasks when compared in the fixed-parameter setting, it is competitive in the *fixed-compute* setting where we measure compute as the amount of TPUv4 hours spent on training, which is a good proxy for the cost of training that model. Furthermore, we also experimented with a hybrid model in which we sparingly interleave Transformer layers (having local attention) in a GSS stack to allow for a richer modeling of short range interactions. To our delight, this further improves performance at (roughly) no extra training cost, both in terms of parameters and compute.

In our experiments we *train* on sequences of length at most 4k, but *evaluate* on a wide range of sequence lengths up to 65k. The performance actually *improves* as the sequence length is increased, suggesting that GSS utilizes extra context despite not being trained with this context. Further, at inference time, state space models including GSS are quite efficient since decoding can happen in recurrent mode (as much as 60× better in the case of S4 (Gu et al., 2022a)). Though, the hybrid model which also uses local attention complicates this advantage a bit.

In summary, we propose GSS, a simple-to-implement alternative to S4 and DSS which trains 2-3× faster, and is competitive with Transformer-based baselines on long-range language modeling benchmarks.

## 2 RELATED WORK

In recent years, attention-based models have emerged as a dominant technique for sequence modeling, achieving remarkable improvements in a wide range of tasks, starting in NLP (Vaswani et al., 2017; Devlin et al., 2019; Radford et al., 2019; Liu et al., 2019), then moving to other classical machine learning areas such as computer vision (Dosovitskiy et al., 2021) and now to the physical sciences (Avsec et al., 2021; Jumper et al., 2021). In brief, an attention layer takes as input three matrices $K, Q, V$ in $\mathbb{R}^{L \times d}$, which should usually be thought of as length $L$ lists of $d$-dimensional vectors. The attention layer then outputs $Y = \sigma(QK^\top)V \in \mathbb{R}^{L \times d}$ where $\sigma$ indicates a row-wise softmax operation. In the popular self-attention variant, $K, Q$ and $V$ are themselves learned functions of a single input sequence $X = (x_1, \ldots, x_L)^\top \in \mathbb{R}^{L \times d}$. Unfortunately, this requires $O(L^2)$ time and space due to the need to construct the matrix $QK^\top \in \mathbb{R}^{L \times L}$.

The surge in popularity of attention has engendered a corresponding surge in interest on methods for increasing the context length $L$ while controlling the $O(L^2)$ computational cost. Broadly speaking, most approaches fall into two camps: those that attempt to "linearize" attention, and those that sparsify the attention matrix $QK^\top$. The first camp exploits the fact that, when softmax is removed, we have $(QK^\top)V = Q(K^\top V)$, where now $K^\top V \in \mathbb{R}^{d \times d}$ and so the whole operation is only linear in $L$ rather than quadratic. This idea is the underlying principle behind methods such as the Performer (Choromanski et al., 2021), Linear Attention (Katharopoulos et al., 2020), Random Feature Attention (Peng et al., 2021) or cosFormer (Qin et al., 2022). In the other camp, one simply sparsifies the matrix $QK^\top$ limit computation and storage requirements. This camp includes BigBird (Zaheer et al., 2020), GMAT (Gupta & Berant, 2020), Longformer (Beltagy et al., 2020), and Blockwise self-attention (Qiu et al., 2020), Reformer (Kitaev et al., 2020) and Perceiver AR (Hawthorne et al., 2022). These approaches can be viewed as a trade-off between performance and computation time (or memory).

Despite its current empirical success, the attention mechanism is not the only approach for modeling sequence data, nor even necessarily the most natural. For example, a classical *recursive* or *state space* layer operates on an input $(x_1, \ldots, x_L)^\top \in \mathbb{R}^{d \times L}$ by defining a sequence of "states" $s_{t+1} = T(s_t, x_{t+1})$ and returning the output sequence $y_t = E(s_t)$ where $T$ and $E$ are learned "transition" and "emission" functions. Assuming $T$ and $E$ can be computed efficiently, this requires only $O(L)$ time in theory. Through the years, many different possibilities for $T$ and $E$ have appeared, such as the simple RNN, which sets $T$ and $E$ to both be MLPs, or more complicated LSTM layer (Hochreiter & Schmidhuber, 1997). Nevertheless, the performance of state space models has in many cases been quickly surpassed by attention.

There have been several recent efforts to recapture the favorable properties of state space models while maintaining or improving upon the performance of attention models, such as the Transformer-XL (Dai et al., 2019), SRU++ (Lei, 2021), or Block-Recurrent Transformers (Hutchins et al., 2022). Of particular relevance to this paper, alternative models based on linear dynamical systems have shown great promise (Gu et al., 2022a; Gupta et al., 2022; Gu et al., 2022b). Rather than attempting to "fix" attention, these are classical state space models where both $T$ and $E$ are linear. An important insight in these works is to recast the model as a convolution with a very large kernel, and then leverage the convolution theorem to compute the sequence $y_t$ in $O(L \log(L))$ time using Fast Fourier Transform (Cooley & Tukey, 1965). While seemingly worse than $O(L)$, this operation is easily parallelizable, and so in practice is significantly faster. Moreover, in a situation in which the $\log(L)$ factor is onerous, one may still fall back to the serial $O(L)$ algorithm. Our approach will build on these works.

## 3 METHOD

We start by reviewing the necessary background on state spaces required to fully describe our model (§3.1). We then formally define our GSS model in §3.2 and the GSS-Transformer-Hybrid in §3.3.

### 3.1 State Space Preliminaries

While in this work we are interested in modeling sequences of vectors, let us first review how state space models define a sequence-to-sequence map for 1-D sequences.

**State Spaces** A discretized state space, parameterized by a state matrix $A \in \mathbb{R}^{N \times N}$, vectors $B \in \mathbb{R}^{N \times 1}$, $C \in \mathbb{R}^{1 \times N}$ and a sample time $\Delta \in \mathbb{R}_{>0}$ defines a sequence-to-sequence map from input $(u_0, \ldots, u_{L-1}) = u \in \mathbb{R}^L$ to output $(y_0, \ldots, y_{L-1}) = y \in \mathbb{R}^L$ via the recurrence[1],

$$
\begin{aligned}
x_k &= \overline{A} x_{k-1} + \overline{B} u_k \quad , \quad y_k = \overline{C} x_k \\
\overline{A} &= e^{A\Delta} \ , \ \overline{B} = (\overline{A} - I) A^{-1} B \ , \ \overline{C} = C \ .
\end{aligned}
\tag{1}
$$

Assuming $x_{-1} = 0$ for simplicity, the above recurrence can be explicitly unrolled as

$$
y_k \ = \ \sum_{j=0}^{k} \overline{CA}^j \overline{B} \cdot u_{k-j} \ .
\tag{2}
$$

For convenience, the scalars $\overline{CA}^k \overline{B}$ are gathered to define the SSM kernel $\overline{K} \in \mathbb{R}^L$ as

$$
\overline{K} \ = \ (\overline{CB}, \overline{CAB}, \ldots, \overline{CA}^{L-1} \overline{B}) \ = \ ( Ce^{A \cdot k\Delta}(e^{A\Delta} - I)A^{-1}B )_{0 \le k < L},
\tag{3}
$$

where the last equality follows by substituting the values of $\overline{A}, \overline{B}, \overline{C}$ from Equation 1. Hence,

$$
y_k \ = \ \sum_{j=0}^{k} \overline{K}_j \cdot u_{k-j} \ .
\tag{4}
$$

where $\overline{K}_j$ denotes the value of the kernel at position $j$. Given an input sequence $u \in \mathbb{R}^L$, it is possible to compute the output $y \in \mathbb{R}^L$ sequentially via the recurrence in Equation 1. While this property is highly desirable for autoregressive decoding, a sequential computation is prohibitively slow to train with long inputs and, instead, Equation 4 can be used to compute all elements of $y$ in parallel, provided we have already computed $\overline{K}$.

**Computing $y$ from $u$ and $\overline{K}$ is easy.** Given an input sequence $u \in \mathbb{R}^L$ and the SSM kernel $\overline{K} \in \mathbb{R}^L$, naively using Equation 4 for computing $y$ would require $O(L^2)$ multiplications. Fortunately, this can be done much more efficiently by observing that for the univariate polynomials

$$
\overline{K}(z) = \sum_{i=0}^{L-1} \overline{K}_i z^i \ \text{ and } \ u(z) = \sum_{i=0}^{L-1} u_i z^i,
$$

$y_k$ is the coefficient of $z^k$ in the polynomial $\overline{K}(z) \cdot u(z)$, i.e. all $y_k$'s can be computed simultaneously by multiplying two degree $L - 1$ polynomials. It is well-known that this can be done in $O(L \log(L))$ time via Fast Fourier Transform (FFT) (Cormen et al., 2009). We denote this fast computation of Equation 4 via the discrete convolution as

$$
y = \overline{K} *_c u \ .
\tag{5}
$$

**Diagonal State Spaces** The challenging part is computing $\overline{K}$ itself as it involves computing $L$ distinct matrix powers (Equation 3). Gupta et al. (2022) observed that the state matrix $A$ can be assumed to be diagonal without loss in performance, thereby allowing a straighforward computation of $\overline{K}$. Their $\text{DSS}_{\text{EXP}}$ model (DSS from hereon) assumes $A$ to be a diagonal matrix $\text{diag}(\lambda_1, \ldots, \lambda_N)$ and assumes $B = (1)_{1 \le i \le N}$. DSS has parameters $\Lambda_{\text{re}}, \Lambda_{\text{im}} \in \mathbb{R}^N$, $C \in \mathbb{C}^N$ and $\Delta_{\log} \in \mathbb{R}$. The diagonal of $A$ (i.e. $(\lambda_1, \ldots, \lambda_N)$) is computed as $-\text{elementwise-exp}(\Lambda_{\text{re}}) + i \cdot \Lambda_{\text{im}}$ where $i = \sqrt{-1}$

---

[1] Discretization used in Equation 1 relies on zero order hold (ZOH) which assumes a constant value between sampled times. We also experimented with other discretization methods such as bilinear transform but did not see much difference in performance.

and $\Delta$ is computed as $\exp(\Delta_{\log}) \in \mathbb{R}_{>0}$. For this parameterization, the kernel (Equation 3) can be computed as a matrix-vector product

$$\overline{K} = (C * ((e^{\lambda_i \Delta} - 1)/\lambda_i)_{1 \le i \le N})_{1 \times N} \cdot \text{elementwise-exp}(P_{N \times L}) \tag{6}$$

where $P_{i,k} = \lambda_i k \Delta$ and $*$ denotes elementwise multiplication.

This formally defines a *linear* sequence-to-sequence map for 1-D sequences. In the case of sequences of $H$-dimensional vectors, state space models are applied individually on the $H$ features as follows.

Each DSS layer receives a length-$L$ sequence $u$ of $H$-dimensional vectors as input, i.e., $u \in \mathbb{R}^{H \times L}$, and produces an output $y \in \mathbb{R}^{H \times L}$. The parameters of the layer are $\Lambda_{\text{re}}, \Lambda_{\text{im}} \in \mathbb{R}^N$, $\Delta_{\log} \in \mathbb{R}^H$ and $C \in \mathbb{C}^{H \times N}$. For each coordinate $h = 1, \ldots, H$, a state space kernel $\overline{K}_h \in \mathbb{R}^L$ is computed for $\Lambda_{\text{re}}, \Lambda_{\text{im}}, (\Delta_{\log})_h, C_h$ via Equation 6. The output $y_h \in \mathbb{R}^L$ for coordinate $h$ is computed from $u_h \in \mathbb{R}^L$ and $\overline{K}_h$ using Equation 5.

For batch size $B$, sequence length $L$ and hidden size $H$, the DSS layer requires $O(NHL)$ time to compute the kernels, and $O(BHL \log(L))$ time for the discrete convolution. In S4 and DSS, the authors suggest to use $N = 64$ as default and for this choice of $N$, the time taken to compute the kernels becomes an important factor, specially for small batches.

## 3.2 GSS LAYER

We build on the idea of the recently introduced Gated Attention Unit (GAU) (Hua et al., 2022) and replace the $\Omega(L^2)$ attention used in GAU by a further simplified DSS layer (§3.1). Gating allows our model to be contextualized over a reduced dimensionality and the use of state spaces provides it with superior contextualizing abilities, while enjoying $O(L \log L)$ complexity out of the box.

Concretely, our *Gated State Space* ("GSS") layer maps a sequence $X \in \mathbb{R}^{L \times E}$ to an output $O \in \mathbb{R}^{L \times E}$ as

$$U = \phi(XW_1) \in \mathbb{R}^{L \times H} \qquad V = \phi(XW_2) \in \mathbb{R}^{L \times F} \tag{7}$$

$$Y = \text{DSS}(U) \in \mathbb{R}^{L \times H} \qquad U_{\text{context}} = YW_3 \in \mathbb{R}^{L \times F} \tag{8}$$

$$O = (U_{\text{context}} * V)W_4 \in \mathbb{R}^{L \times E} \tag{9}$$

where $*$ denotes elementwise multiplication and $\phi$ denotes GELU (Hendrycks & Gimpel, 2016). Note that, contrary to Hua et al. (2022), in our experiments, we did not observe much benefit from using $\text{RELU}^2$ or Swish activations instead of GELU.

In Equation 8, a contextualized representation of $U$ is formed via DSS in $O(NHL) + O(BHL \log L)$ time. In most of our experiments we used $H = E/4 = 256$, $N = 512$ and $F = 4096$. Similar to GAU, gating allows us to perform a weaker contextualization using the state space layer by reducing the dimension of the input to DSS , by $4\times$ in our case. This offers a much needed increase in throughput since we observed that FFT ops were the main bottleneck (measured on TPUs). As shown in Table 1, this provides more than $3\times$ speedups over the vanilla DSS block described in §3.1.

**Simplified DSS used in GSS layer** In Equation 8, instead of directly using DSS as described in §3.1, we made several key simplifications based on our exploratory results.

As described in §3.1, S4 and DSS train a separate $\Delta$ for each of the $H$ coordinates resulting in the creation of $H$ separate $P$ matrices in Equation 6. In the DSS layer used in GSS, we decided to eliminate the use of $\Delta$ by fixing it as 1. This reduces the compute required for creating the kernels and makes the kernel computation simpler.

Secondly, we found that randomly initializing the parameters corresponding to $\Lambda$ works just as well as the Skew-Hippo initialization suggested in (Gupta et al., 2022). Importantly, we parametrize both real and imaginary parts of $\Lambda$ in log space, so that the random initializations span many orders of magnitude (§A.2). The effectiveness of random initialization is in contrast to the findings of Gu et al. (2022a) and Gupta et al. (2022) who reported their respective initializations of the state space parameters to be critical to the model performance. We do however note that the experiments in

our setting of large-scale language modeling are conducted on orders of magnitude larger scale of compute than what is used in the tasks considered in these works. Moreover, the data modalities we consider in this work are different from the tasks considered in these works (e.g. Long Range Arena) which are specifically designed to stress test long-range reasoning.

### 3.3 GSS-Transformer-Hybrid

Conceptually, GSS looks fairly different from the current workhorse of machine learning; the Transformer architecture. Given this, it is not immediately clear if one is decidedly better than the other or if they each provide some orthogonal benefits. If its the latter, one might wonder if there are synergies between these architectures which can be exploited to create a hybrid model which is stronger than either one of them individually. To that end, we also consider a conceptually simple hybrid between GSS and Transformer where we sparingly interleave traditional Transformer blocks with GSS layers. Despite its glaring simplicity, as shown in Table 2, we observed that it shows consistent and significant improvements on all tasks.

**Chunking long inputs**  In all our experiments we used sequence lengths large enough to be prohibitive for traditional Transformer layers. To get around this restriction at the Transformer layers used in our hybrid model, we chunk their inputs into non-overlapping chunks of length 512 and run the Transformer layer on each of them independently.

## 4    Results

We conduct experiments with GSS on 4 different datasets, LM1B, PG19, ArXiv and Github, each of them varying qualitatively with another in terms of modality and average document length.

**LM1B** is a standard and reasonably big dataset (1B tokens) where each training example consists of a short sentence (Chelba et al., 2014). This is different from rest of the datasets which consists of a much larger sequence of tokens per example. Although, our primary goal is to measure GSS's ability to capture long range dependencies, we include results on LM1B benchmark and compare with vanilla Transformer baseline, which is hard to do for larger sequence lengths.

**PG19** dataset is constructed from extracting a large collection of full-length books from Project Gutenberg (Rae et al., 2020). All extracted books were written before 1919 and only contain tokens from the English language. Over the years PG-19 has become a standard benchmark for measuring progress on long range dependency modeling over text.

**ArXiv Math** dataset was recently collected by Wu et al. (2022) and contains LaTeX source for articles focusing on Mathematics. Even though articles are typically shorter than full-length books, as in PG19, since LaTeX source can have many special characters, typical sub-piece vocabularies are not very effective and tends to produce examples of similar size, as measured by number of tokens. It is possible to train a custom sub-piece vocabulary for this dataset, but we stick with the vocabulary used by both Wu et al. (2022) and Hutchins et al. (2022) for fair comparison.

**Github** was also first collected and used by Wu et al. (2022). It is a corpus of raw source code collected from several Github repositories with open-source licences. The dataset contains code from several programming languages, including C, C++, Java, Python, Go and Typescript. In this case individual files can be small but code repositories are typically is organized so that code can be reused across file boundaries. Thus, for every repository, all the files were concatenated to produce a single document.

### 4.1    Comparing DSS and GSS

As shown in Table 1, in our first set of results, we compare DSS and GSS models both in terms of perplexity and throughput, as measure by steps per second on all 4 datasets. For the PG-19, Arxiv and Github datasets, all models were trained with 4k sequence length at training time. However, at evaluation time, we present results for a large range of sequence lengths. Despite being a (slightly) smaller model, GSS improves perplexity while being 2-3× faster depending on these datasets. In contrast, for LM1B, we train and evaluate with sequence length of 512, since the dataset only contains short sentences, most examples comfortably fully fitting in the context window.

| Dataset | Model | Params | Throughput (steps/sec) | Eval Sequence Length | | | |
|---------|-------|--------|------------------------|--------------------|---|---|---|
| | | | | Perplexity | | | |
| | | | | 512 | 4k | 16k | 65k |
| LM1B | Transformer | 182M | 6.6 | 13.51 | | | |
| | DSS | 188M | 1.8 | 13.59 | | | |
| | GSS | 190M | 5.6 | 13.26 | | | |
| PG-19 | Transformer | 182M | 6.6 | 13.00 | | | |
| | DSS | 209M | 1.8 | 14.51 | 13.53 | 692.1 | OOM |
| | GSS | 192M | 5.3 | 14.01 | 12.84 | 12.94 | 12.47 |
| Arxiv | Transformer | 182M | 6.6 | 3.27 | | | |
| | DSS | 209M | 1.8 | 3.65 | 3.13 | 284.6 | OOM |
| | GSS | 192M | 5.3 | 3.57 | 3.08 | 3.08 | 2.75 |
| Github | Transformer | 182M | 6.6 | 2.88 | | | |
| | DSS | 209M | 1.8 | 3.65 | 3.21 | 242.7 | OOM |
| | GSS | 192M | 5.3 | 2.68 | 2.35 | 2.31 | 2.12 |

Table 1: Comparison of DSS and GSS models in fixed-param setting. We consistently find that GSS outperforms DSS (with hyperparameters taken from Gupta et al. (2022)) while being 2-3× faster on all tasks. We train both GSS and DSS with sequence length 4k except for LM1B. Training sequence length is 512 for the Transformer baseline for all datasets.

Furthermore, we observe significant generalization over changes in sequence length on all the datasets. Not only does the performance not degrade when increasing sequence length, it actually improves quite a bit! This suggests that the model is effective at utilizing the extra context even though the model was not trained with that amount of context. Note that we used default hyperparameters suggested by Gupta et al. (2022) for initializing state space variables for DSS. But, for GSS, since we made several structural changes, we retuned the hyperparameters related to initialization on PG19 alone and used them for all the datasets. Since we had access to evaluation metrics for all sequence lengths, length generalization factored in for our choice of hyperparameters, which is not true for DSS. It may be possible to see similar length generalization even with vanilla DSS if initialization hyperparameters were retuned.

Similar to language modeling experiments in (Gu et al., 2022a), every block of DSS baseline consists of DSS layer followed by GLU (Dauphin et al., 2017) and a Feedforward layer similar to the one used in Transformer block with GELU activation (Hendrycks & Gimpel, 2016). DSS baseline consists 12 layers and an embedding dimension of 1024. For GSS, we increased the number of layers to 16 to match the parameter count.

## 4.2 Comparison with other baselines

In this section, we turn our attention towards apples-to-apples comparison of GSS versus well-tuned baselines on these datasets. For a complete picture of the cost of training these models, we report both the number of parameters and time spent training as measured by TPUv4 hours. For baselines, we selected the best performing models reported in (Hutchins et al., 2022) for every dataset and compare with GSS model both in fixed-param and fixed-compute settings.

**GSS models** GSS consists of 16 layers and an embedding dimension of 1024. We also consider a larger variant with 32 layers as denoted by GSS-L. For GSS-Hybrid model, we used vanilla Transformer blocks at every 4th layer starting with the 2nd layer. Since GSS layers are inherently position aware, using them for the 1st layer eschews any need of explicit position embeddings typically used with otherwise position invariant Transformer blocks. Thus, barring position aware nature of GSS layers, we don't use any kind of explicit position embedding or bias in our models. For the Transformer blocks used in hybrid models, we use multi-head self-attention with 8 heads, each with size 128.

| Dataset | Model | Params | TPUv4 hours | Eval Sequence Length | | | |
|---|---|---|---|---|---|---|---|
| | | | | Perplexity | | | |
| | | | | 512 | 4k | 16k | 65k |
| PG-19 | Rec:fixed:skip | 196M | 0.8k | | 11.55 | | |
| | Feedback:lstm:single | 196M+ | 0.8k+ | | 11.31 | | |
| | Feedback:fixed:skip | 196M+ | 0.8k+ | | 11.24 | | |
| | **GSS** (**this work**) | 192M | 0.5k | 14.01 | 12.84 | 12.94 | 12.47 |
| | **GSS**-L | 352M | 0.8k | 12.48 | 11.33 | 11.16 | 11.12 |
| | **GSS**-Hybrid-L | 373M | 0.8k | 11.45 | **10.52** | 10.44 | 10.1 |
| Arxiv | Rec:fixed:skip | 196M | 0.8k | | 2.36 | | |
| | Feedback:lstm:single | 196M+ | 0.8k+ | | **2.33** | | |
| | Feedback:fixed:skip | 196M+ | 0.8k+ | | 2.36 | | |
| | GSS | 192M | 0.5k | 3.57 | 3.08 | 3.08 | 2.75 |
| | GSS-L | 352M | 0.8k | 3.29 | 2.71 | 2.72 | 2.51 |
| | GSS-Hybrid-L | 373M | 0.8k | 2.94 | 2.51 | 18.27 | 125.2 |
| Github | Rec:fixed:skip | 196M | 3k | | 2.04 | | |
| | Feedback:lstm:single | 196M+ | 3k+ | | 2.07 | | |
| | Feedback:fixed:skip | 196M+ | 3k+ | | 2.16 | | |
| | GSS | 192M | 0.5k | 2.68 | 2.35 | 2.31 | 2.12 |
| | GSS-L | 352M | 1.8k | 2.31 | 1.99 | 2.20 | 2.28 |
| | GSS-Hybrid-L | 373M | 1.8k | 2.34 | **1.88** | 1.74 | 2.09 |

Table 2: Comparing GSS variants with best-performing models from Hutchins et al. (2022) in both fixed-param and fixed-compute settings. While, in the fixed-param setting, GSS currently lags behind performance of best Block Recurrent Transformer baseline, it is fairly competitive in the fixed-compute setting (as measured by total TPUv4 hours taken to complete training). For the larger model GSS-L used for fixed compute comparison, we simply double the layers from 16 to 32 keeping everything else fixed. In addition, due to the inherent recurrent view of state space model family, decoding at inference time is much faster than Transformer based baselines. For block-recurrent baselines, adding feedback increases both parameter count and training time, we stick with conservative estimates derived from Hutchins et al. (2022) for both, which we denote by '+'. Hutchins et al. (2022) report $\log_2$ of perplexity which we convert to raw perplexity in this table. Param count for all the models include embedding layers as well. A comparison with FLASH (Hua et al., 2022) is included in Appendix C (Table 4).

**Baselines** We considered 3 high-performing baselines and numbers reported in (Hutchins et al., 2022). Block Recurrent Transformer leverages recurrence over blocks of Transformer layer to model long range dependencies. Hutchins et al. (2022) performed a comprehensive exploration of open design space of incorporating recurrence across blocks. Somewhat surprisingly, Rec:fixed:skip, which accumulates recurrent state vector as an exponential moving average over time performs better than more complicated gating designs. Another variation which performed well with Block Recurrence is the idea of adding a feedback mechanism over blocks similar to Feedback Transformer (Fan et al., 2020). Note that feedback mechanism makes training more expensive due to additional cross-attention modules and corresponding paramaters.

**Fixed-param comparison** As shown in Table 2, we see that GSS variants come very close but not quite beat the strongest block recurrent baseline in the fixed-param setting. In this case, we are comparing GSS model which has roughly 192M parameters (including embeddings) with the baselines all of which have around 196M parameters. Even though the parameter count is fixed, we see that GSS runs faster than block recurrent baselines, likely due to the fact that all the layers can be completely parallelized unlike the the recurrence in the baselines which run in a sequential fashion.

**Fixed-compute comparison** Since the GSS model runs faster than the baselines, we also train with versions larger than GSS such that the training time (as measured by total TPUv4 hours) matches the time taken by the baselines. We simply double the number of layers from 16 to 32 to construct GSS-L. As expected, adding more parameters improves perplexity numbers on the eval set of all the datasets. Moreover, we find that the GSS-Hybrid versions of the model outperform the best baseline

| Model | Params | Layers | WLP | Vocabulary |
|---|---|---|---|---|
| Transformer-XL (Rae et al., 2020) | | 36 | 36.3 | 32k |
| Compressive Transformer (Rae et al., 2020) | | 36 | 33.6 | 32k |
| Routing Transformer (Roy et al., 2021) | 490M | 22 | 33.2 | 98k |
| Perceiver AR (Hawthorne et al., 2022) | 974.6M | 60 | 28.9 | 32k |
| Block Recurrent Trans (Hutchins et al., 2022) | 1.3B | 24 | 26.5 | 32k |
| GSS-Hybrid (ours) | 341M | 32 | 32.9 | 32k |
| | 681M | 64 | 30.2 | 32k |
| | 1B | 96 | 28.0 | 32k |
| | 1.4B | 128 | 27.3 | 32k |

Table 3: Comparison of GSS with other competitive models on PG-19 across multiple scales as measured by word-level perplexity (WLP) on test set of PG-19. We evaluated WLP on 16k sequence length for GSS-Hybrid set of models. Model size numbers for Perceiver AR (Hawthorne et al., 2022) and Routing Transformer (Roy et al., 2021) are taken from Hutchins et al. (2022).

model on PG-19 and Github datasets. We do see significant improvements for Arxiv dataset as well but unfortunately not enough to be stronger than the baseline. We think this may be resolved by the use of a vocabulary more suited to Arxiv symbols. On the flip side, we can no longer do a fair comparison with token level perplexity if we change the vocabulary, so we stick with the vocabularies used by the baselines for this study.

**Length generalization** We train all variants of GSS with sequence length of 4k but evaluate on 4 different lengths $l \in [512, 4k, 16k, 65k]$. On PG-19, we see significant length generalization across the board. Not only the performance doesn't degrade as the sequence length is increased but it gets significantly better for all model variants. On Arxiv and Github, the situation is a little more complicated. For smaller models, we still see length generalization but it tends to degrade when either the model is made bigger or the dataset has a lot of noise (as indicated by variation in perplexity metric over time). How to robustly achieve length generalization is an interesting research question on it is own and we believe one can design interventions which can lead to further improvements, which we don't explore here.

Note that block recurrent baselines, with or without feedback mechanism, process documents in a sequential fashion such that recurrent state from previous segment from the document is passed to the next segment, with backward pass truncated to the current segment. This means that, even though the segment sequence length is set to 4k, block recurrent models have (arguably weak) access to almost the entire past. Thus, perplexity comparison at sequence length 4k is slightly unfair towards GSS models since they do *not* employ such state caching mechanisms.

## 4.3 SCALING UP

In this section, we further scale up the GSS model focusing only on PG19 dataset. As shown in Table 3, even though GSS does not beat current state of the art on a fixed-param basis, we consistently observe significant gains from simply scaling up, similar to popular Transformer based language models like GPT-3 (Brown et al., 2020) and PaLM (Chowdhery et al., 2022). Furthermore, GSS also outperforms Routing Transformer (Roy et al., 2021) and Perceiver AR (Hawthorne et al., 2022) when comparing similar sized models.

**CONCLUSION** We introduce GSS, a general purpose sequence model which leverages gated units and trains significantly faster as shown on several language modeling benchmarks. Further comparison with well tuned Transformer baselines suggests that GSS is fairly competitive in fixed-compute setting. We further show that a hybrid model constructed by interleaving GSS and Transformer improves performance even further. We however recognize that perplexity numbers alone are not satisfying for gauging overall model performance. As part of future work, we will explore scaling GSS even further and conducting few-shot evaluation on standardized benchmarks, comparing it against other popular large language models.

## ACKNOWLEDGMENTS

We are grateful to the developers of Jax and Flax libraries. In addition, we would like to thank DeLesley Hutchins and Imanol Schlag for answering our questions, sharing the datasets and helping us in general with details of Block Recurrent Transformer paper. Finally, we are grateful to Ethan Dyer for discussions on long context modeling, Walid Krichene, Albert Gu, John Anderson and anonymous reviewers for providing feedback on an earlier draft of this paper. Ashok Cutkosky acknowledges support from NSF grant CCF-2211718.

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

# A  SUPPLEMENTAL MATERIAL

## A.1  FAST CONVOLUTION VIA FFT

For $u, v \in \mathbb{C}^{1 \times L}$ the Circular Convolution Theorem states that,

$$\mathrm{invFFT}_L(\mathrm{FFT}_L(u) * \mathrm{FFT}_L(v)) = v \cdot \begin{bmatrix} u_0 & u_1 & \cdots & u_{L-1} \\ u_{L-1} & u_0 & \ddots & \vdots \\ \vdots & \ddots & \ddots & u_1 \\ u_1 & \cdots & u_{L-1} & u_0 \end{bmatrix} = v \cdot \mathrm{circulant}(u) \,.$$

where $*$ denotes elementwise multiplication. As $\mathrm{FFT}, \mathrm{invFFT}$ can be done in $O(L \log L)$ time this provides a fast algorithm for circulant matrix-vector product. In practice, linear systems can often be expressed as a circulant matrix-vector product and is also true in the case of Equation 4 which can be equivalently expressed as

$$[y_0 \ \ldots \ y_{L-1} \ | \ \ldots \ ]_{1 \times 2L} = [\overline{K} \ | \ 0 \ \ldots \ 0]_{1 \times 2L} \cdot \mathrm{circulant}([u_0 \ \ldots \ u_{L-1} \ | \ 0 \ \ldots \ 0])_{2L \times 2L} \,.$$

## A.2  IMPLEMENTATION OF GSS

```python
def log_initializer(min, max):

  def init(shape):
    return random.uniform(shape) * (log(max) - log(min)) + log(min)

  return init

def simplified_dss_kernel(H, L, N=512):
    # Lambda_re, Lambda_im: [N]
    # C_re, C_im: [H N]
    Lambda = -Lambda_re.exp() + 1j*Lambda_im.exp()   # [N]
    C = C_re + 1j*C_im                               # [H N]
    S = (Lambda * arange(L).view(1,L)).exp()         # [N L]
    C = C * (Lambda.exp() - 1) / Lambda              # [H N]
    return einsum('hn,nl->hl', C, S).real            # [H L]

def dss(u, H, L):
    u = norm(u)
    # compute H state space kernels
    K = simplified_dss_kernel(H, L)
    K_f = rfft(K, pad_to=2*L)
    u_f = rfft(u, pad_to=2*L)
    y = irfft(K_f * u_f)[...,:L]
    # param D: [H,1]
    return y + D * u

def gss(x, F=4096, L=4096, E=1024, H=256):
    shortcut, x = x, norm(x)
    v = dense(x, F, activation='gelu')
    u = dense(x, H, activation='gelu')
    y = dss(u, H, L)
    uc = dense(y, F)
    o = dense(uc * v, E)
    return o + shortcut
```

Figure 2: Pseudocode of GSS (§3.2).

Lambda_re and Lambda_im are initialized with log_initializer set to (min=1e-3, max=1e0) and (min=1e-5, max=1e2) respectively. We tuned these parameters on PG-19 on 4k sequence length and used them for all our experiments. We recommend re-tuning these if the training sequence length is changed.

| Dataset | Model | Params | TPUv4 hours | Eval Sequence Length | | | |
|---------|-------|--------|-------------|-------------|------|------|------|
| | | | | Perplexity | | | |
| | | | | 512 | 4k | 16k | 65k |
| PG-19 | FLASH (ours) | 340M | 1k | 12.88 | 11.42 | | |
| | GSS-L | 352M | 0.8k | 12.48 | 11.33 | 11.16 | 11.12 |
| | GSS-Hybrid-L | 373M | 0.8k | 11.45 | **10.52** | 10.44 | 10.1 |
| Arxiv | FLASH (ours) | 340M | 1k | 3.22 | **2.48** | | |
| | GSS-L | 352M | 0.8k | 3.29 | 2.71 | 2.72 | 2.51 |
| | GSS-Hybrid-L | 373M | 0.8k | 2.94 | 2.51 | 18.27 | 125.2 |
| Github | FLASH (ours) | 340M | 2.5k | 2.65 | 1.90 | | |
| | GSS-L | 352M | 1.8k | 2.31 | 1.99 | 2.20 | 2.28 |
| | GSS-Hybrid-L | 373M | 1.8k | 2.34 | **1.88** | 1.74 | 2.09 |

Table 4: Comparing GSS variants with our re-implementation of the best-performing models from Hua et al. (2022) in both fixed-param and fixed-compute settings. Param count for all the models include embedding layers as well.

## B    DATA DETAILS

We do token level modeling on all the datasets and report resulting perplexity numbers on a heldout set of examples. Perplexity numbers are obtained using teacher forcing (or parallel mode) where the correct output from the heldout set is used for decoding the next token at each position. For a fair comparison with the baselines we considered, we keep the vocabularies consistent as used by the baselines models. Specifically, we used custom trained 30k sized sentence-piece vocab for LM1B, T5 vocab with 32k tokens for PG19 (Raffel et al., 2020) and Meena vocab with 32k tokens (Adiwardana et al., 2020) for both ArXiv and Github datasets.

## C    COMPARISON WITH FLASH

In this section we compare GSS with similar sized (both param and compute) FLASH baseline (Hua et al., 2022) which also incorporates gated activation. As shown in Table 4, we see a trend similar to the comparison with Block Recurrent Transformer where GSS-Hybrid outperforms FLASH baseline on PG-19 and Github but lags behind on Arxiv dataset. In addition, we see that GSS-L is slightly more compute efficient than FLASH since it achieves better performance of PG-19 and Github while requiring less compute.

**FLASH hyperparams.** We re-implemented FLASH baseline in our setup while being faithful to the architecture and hyperparameters reported in Hua et al. (2022). In order to do a fair comparison we made small number of changes, 1) we set the number of layers in FLASH to 32, which matches the GSS baseline and 2) we increase "expansion rate" from 2 to 3 to match the parameter count of GSS. Other than that, we repurposed all well-tuned hyperparameters as is from Hua et al. (2022), except for the learning rate which we increased linearly, proportional to the number of tokens per batch. More precisely, Hua et al. (2022) set learning rate to 0.0007 with $2^{18}$ tokens per batch. Since GSS was trained with $2^{19}$ tokens per batch, in order to be fair we doubled the learning rate to 0.0014 for the FLASH baseline for PG-19 and Arxiv datasets. Although, due to a lot of noise in the Github dataset, we stick with the lower learning rate of 0.0007 since a high learning rate of 0.0014 led to the model diverging half way through the training run.

## D    ADDITIONAL ANALYSIS

In this section, we perform two ablations on GSS and GSS-Hybrid.

**Varying kernel length.** In Table 5, we report how the performance of GSS and GSS-Hybrid changes as the length of kernel used in our simplified DSS is varied. In all of our experiments so far, we

| Model | Kernel Length | Params | TPUv4 hours | Eval Sequence Length | | | |
|---|---|---|---|---|---|---|---|
| | | | | Perplexity | | | |
| | | | | 512 | 4k | 16k | 65k |
| GSS | 64 | 192M | 0.52k | 14.81 | 14.00 | 14.21 | 13.56 |
| | 256 | 192M | 0.49k | 14.29 | 13.36 | 13.36 | 12.95 |
| | 1024 | 192M | 0.48k | 14.22 | 13.10 | 12.80 | 13.45 |
| | 4096 | 192M | 0.55k | 14.01 | 12.84 | 12.94 | 12.47 |
| GSS-Hybrid | 64 | 203M | 0.55k | 12.87 | 12.90 | 12.01 | 12.09 |
| | 256 | 203M | 0.52k | 12.90 | 12.70 | 11.88 | 11.96 |
| | 1024 | 203M | 0.55k | 12.90 | 11.86 | 11.73 | 11.41 |
| | 4096 | 203M | 0.56k | 12.82 | 11.78 | 11.66 | 11.70 |

Table 5: Comparing GSS when kernel length is varied on PG-19.

| Model | Attention Length | Params | TPUv4 hours | Eval Sequence Length | | | |
|---|---|---|---|---|---|---|---|
| | | | | Perplexity | | | |
| | | | | 512 | 4k | 16k | 65k |
| GSS | N/A | 192M | 0.55k | 14.01 | 12.84 | 12.94 | 12.47 |
| GSS-Hybrid | 64 | 203M | 0.54k | 13.66 | 12.67 | 12.22 | 12.19 |
| | 128 | 203M | 0.51k | 13.44 | 12.35 | 12.00 | 12.02 |
| | 256 | 203M | 0.54k | 13.15 | 12.09 | 12.11 | 11.99 |
| | 512 | 203M | 0.56k | 12.82 | 11.78 | 11.66 | 11.70 |

Table 6: Comparing GSS when attention chunk length is varied on PG-19.

set the kernel length to be the same as input sequence length. This means that the state space layer accumulates information from all of the past tokens. Intuitively, decreasing the kernel length shortens the context window to past tokens but should train faster. As shown in Table 5, we see that both GSS and GSS-Hybrid trains faster as we decrease context length (barring kernel length of 64, which we believe suffers from padding issues on TPUs since it may be smaller than what can be used in a single instruction). Another thing to note is that, GSS-Hybrid seems to degrade much less than vanilla GSS as kernel length is decreased. For instance, as kernel length is decreased from 4k to 1k, GSS-Hybrid degrades by 0.08 points but GSS degrades by 0.26 points. This suggests that, intermediate Transformer layers may be helping with modeling context lengths beyond the chunk length of 512 that it is limited to. We leave more thorough exploration of this interesting direction to future work.

**Varying attention chunk length.** To fully understand the properties of intermediate attention layers in GSS-Hybrid, we additionally ablate the attention chunk length. As shown in Table 6, as expected, performance significantly degrades as the chunk length is decreased and the values smoothly interpolates between chunk size of 512 and vanilla GSS model. It is interesting to see that even a modest chunk size of 64 leads to better performance than vanilla GSS model.

## E    ADDITIONAL COMPARISON OF DSS VS GSS

In order to do a fairer comparison between DSS and GSS, we ran an additional study as shown in Table 7 where we report numbers for an additional GSS model for which we reduced the number of layers from 16 to 12 and increased the MLP size from 4096 to 6144 in order roughly match the parameter count. We find that GSS-12L is faster yet almost as competitive as GSS (16 layers) on all 3 datasets.

| Dataset | Model | Params | Throughput (steps/sec) | Eval Sequence Length | | | |
|---------|-------|--------|------------------------|------|------|------|------|
| | | | | Perplexity | | | |
| | | | | 512 | 4k | 16k | 65k |
| PG-19 | DSS | 209M | 1.8 | 14.51 | 13.53 | 692.1 | OOM |
| | GSS | 192M | 5.3 | 14.01 | 12.84 | 12.94 | 12.47 |
| | GSS-12L | 209M | 5.7 | 14.07 | 12.96 | 12.71 | 12.36 |
| Arxiv | DSS | 209M | 1.8 | 3.65 | 3.13 | 284.6 | OOM |
| | GSS | 192M | 5.3 | 3.57 | 3.08 | 3.08 | 2.75 |
| | GSS-12L | 209M | 5.7 | 3.57 | 3.06 | 3.06 | 2.8 |
| Github | DSS | 209M | 1.8 | 3.65 | 3.21 | 242.7 | OOM |
| | GSS | 192M | 5.3 | 2.68 | 2.35 | 2.31 | 2.12 |
| | GSS-12L | 209M | 5.7 | 2.67 | 2.33 | 2.23 | 2.33 |

Table 7: Comparison of DSS, GSS and GSS-12L.

## F  TRAINING DETAILS

### F.1  TRAINING DETAILS FOR RESULTS IN TABLE 1

All the models in Table 1 were trained with $2^{19}$ tokens per batch and 125k total training steps. We make sure to change the batch size as a function of the sequence length so that number of tokens in the batch remains the same. For example, for LM1B we set batch size to 1024 and sequence length to 512 but for rest of the datasets we use batch size of 128 and sequence length of 4k. For datasets with longer documents, we also considered increasing the sequence length even further. Intuitively, training on longer sequences would help the model learn longer range dependencies better. On the flip side, it makes the optimization a bit more challenging due to large number of correlated tokens per batch and would even likely result in overfitting. Since GSS is able to generalize beyond the length it was trained on in most cases, we found sequence length of 4k to be a reasonable middle ground for this set of experiments.

### F.2  TRAINING DETAILS FOR RESULTS IN TABLE 2

Similar to Section 4.3, unless otherwise mentioned, all the non-baseline models were trained using 64 TPUv4 cores for 125k steps. We use batch size of 128 and 4k sequence length at training time, with a total of $2^{19}$ tokens per batch. We increase the batch size to 256 for the Github dataset (with a token count of $2^{20}$ per batch) since we observed a lot of noise in our metrics. This is consistent with observations made by Hutchins et al. (2022).

We used Adam optimizer (Kingma & Ba, 2015) and tuned the base learning rate over a grid of $\in$ [0.0064, 0.0032, 0.0016, 0.0008]. We also employ linear warmup for 1k steps and cosine decay until 1e-6. We also observed better performance by using a higher than typical weight decay rate of 0.1, other than that we did not employ any additional regularization techniques, including dropout. Note that similar to (Gu et al., 2022a; Gupta et al., 2022), we used a constant learning rate of 1e-3 and set weight decay rate to 0.0 for state space parameters part of GSS Layer . In addition, we clip the gradient to norm 1.0 before passing to the optimizer. Both of these helped with certain instabilities we observed in our preliminary experiments.

### F.3  TRAINING DETAILS FOR RESULTS IN TABLE 3

For these experiments, we build on our best performing model from Section 4.2 on PG-19 i.e. GSS-Hybrid-L. In order to scale up GSS, we simply increase the number of layers in the model. We also increased the batch size to 256 (from 128) while using 4k sequence length at training time, with a total of $2^{20}$ tokens per batch, keeping everything else exactly the same.

