# OpenReview forum: "Long Range Language Modeling via Gated State Spaces"
_ICLR.cc/2023/Conference — ICLR 2023 poster_

### Official Review · Reviewer_Rso2 · 2022-10-24

**Confidence:** 4
**Correctness:** 3
**Technical Novelty And Significance:** 2
**Empirical Novelty And Significance:** 2
**Recommendation:** 6

**Clarity, Quality, Novelty And Reproducibility:**

The experimental setup in this paper make it difficult to fairly compare GSS with previous models:

1. As the authors created their own vocabularies, the results in Table 3 are not directly comparable.

2. Since in GSS the FFN module is omitted, the authors increases the number of layers to match the number of parameters with DSS. However, we know that deeper neural networks usually deliver better results than shallower ones with higher-hidden dimensions. Have you tried to increases the model dimension instead of increasing the number of layers? I believe it is a more fair comparison.

3. As GSS is directly inspired from GAU and FLASH (Hua, et al, 2022), why the authors have not compared GSS directly with FLASH, but using other variants of Transformers as baselines? Since GSS has very different organization of parameters (more layers and no FFNs), the comparison with FLASH (with the same number of layers and similar model size) would make a more clear position of GSS among previous models, particularly attention-based models.

Other questions about experiments:

1. In GSS, the $N$ is set to 512, while in S4 and DSS, $N$ is usually much smaller, e.g. (64). In the experiments, what is the value of $N$ in DSS? Is it important to use larger $N$?

2. GSS-Hybrid obtained better PPL than GSS, and the authors claimed that the self-attention is modeling local dependencies and the state space model is modeling long-term ones. However, there were no analysis or results to support this. Is it more reasonable that the self-attention is more using to model long-term dependencies (within a chunk of size 512) while the state space model is for local ones across successive chunks?

**Strength And Weaknesses:**

Strengths:
The design of GSS is well-motivated and the paper writing is clear and easy to follow. The authors also clearly claimed their contributions.

Weaknesses:
The main concerns are from experiments. I found that the experimental setup makes the comparisons unfair, which makes it difficult to position GSS among other neural models in language modeling. I elaborated my concerns in the following questions.


**Summary Of The Paper:**

This paper proposed to leverage the gating structure in gated attention unit (GAU) in state space models, named Gated State Space (GSS). By reducing the dimension of the state space module in GSS which requires FFT to compute the output, GSS is faster than the diagonal state space (DSS) model.

Experiments were conducted on four language modeling benchmarks, and the authors demonstrated the effectiveness and efficiency of GSS comparing with DSS and other Transformer-based baselines.

**Summary Of The Review:**

To sum up, I am worried that the paper needs more experiments and analysis before publication.

---

> ### Author Response · Authors · 2022-11-13
> **Response to Reviewer Rso2**
>
> We thank the reviewer for the detailed review and insightful comments! To address your main concerns regarding fair comparison with the baseline, we have provided additional clarification where necessary and also ran additional experiments as you suggested. We hope that our response alleviates your reservations and would consider accepting our work.
>
> Our response to the raised concerns are as follows
> 1. ```vocabularies in Table 3``` : You are right that vocabularies in Table 3 are different across baselines but we would like to point out that PG-19 is an open vocabulary benchmark and we (and other baselines) report **Word Level Perplexity** making the comparison fair across vocabularies. Finally, all the papers for baselines we report in Table 3 also followed the same protocol.
>
> 2. ```increasing depth vs width```: we would like to make a slight correction in our calculation of **model depth**  in the DSS baseline. For DSS, similar to S4, each block contains an additional FFN layer, thus the model depth is in fact 24 instead of 12, while the GSS model still has 16 gated contextualization layers. Note that this way of calculating model depth is also adopted in GAU and FLASH (Hua, et al, 2022). We hope that this alleviates your concern regarding depth.
>
> 3. ```GAU / FLASH baseline```: This is a fair point and one which other reviewers (and even we) share! To that end, we **implemented FLASH baseline** in our setup and compared it with GSS on all 3 long range datasets we considered. The results are presented in Appendix Section B, Table 4. We strived hard to make the comparison fair and found that similar to the comparison with Block Recurrent Transformer, GSS (and GSS-Hybrid) is very competitive with FLASH, where the **GSS-Hybrid model outperforms FLASH on PG-19 and Github**. Additionally, since we used vanilla Transformer blocks in GSS-Hybrid, we believe replacing the Transformer blocks with GAU can further improve performance, but due to resource constraints, we leave this for future work.
>
> 4. ```value of N``` : As clarified in the **Simplified DSS used in GSS layer** section (Page 5), GSS assumes $\Delta = 1$ and hence the same $P$ matrix of size NxL is used for all the $H$ kernels (Equation 6). This is the reason we use a larger $N=512$. Note that even for the DSS baseline we use larger $N=512$. Original DSS paper uses a different $\Delta$ for each of the $H$ dimensions and hence forms $H$ distinct $P$ matrices of size N x L and therefore uses a smaller $N=64$.
>
> 5. ```self-attention to model long-term dependencies (within a chunk of size 512)``` : First, we would like to clarify that, by “local“ we mean lengths at range less than 512 and by “global” we mean lengths beyond that i.e. lengths at which attention becomes prohibitively slow. We apologize for any confusion.
> Second, this is in fact a great suggestion! While it is hard to tease out which component helps model larger context dependencies, your suggestion led us to do a more comprehensive analysis of the kernel length we used and even attention chunk length in the hybrid model. We have **added an analysis section** (Appendix C) in which we train with 1) shorter attention chunks but full state space kernels and 2) vary state space kernel length in both GSS and GSS-Hybrid.
> From ablating **kernel length**, we find that hybrid version leads to less degradation in performance as kernel length is made smaller. Contrary to what we thought, this suggests that intermediate Transformer layers may be helping with modeling context lengths beyond the chunk length of 512 that it is limited to.
> Additionally, in GSS-Hybrid model, we ablate **attention chunk length**, where we find that even a very small chunk length of 64 leads to better performance than vanilla GSS model, suggesting that even a short context attention can provide complementary benefit to GSS.

---

> > ### Comment · Reviewer_Rso2 · 2022-11-15
> > **Response**
> >
> > Thanks for your efforts on these additional experiments. They addressed most of my concerns. The comparison between GSS and Flush provided a clearer position of GSS on language modeling. In addition, the ablation study on chunk length vs. kernel size illustrates the importance of attention on long-term dependency modeling.
> >
> > Therefore, I upgraded my score to 6.
> >
> > I still have the following concern:
> >
> > `depth vs width`. Personally speaking, I do not agree with Hua et al., (2022) on treating Attention and FFN as two separate layers. They should be treat as a single unit, because FFN does not model any contextual information. In this case, your baseline DSS contains `12` ***contextualized layers*** while your GSS has `16`. This is why I believe it is better to reduce your contextual layers to `12` by increasing the width of each layer to keep similar model size.

---

> > > ### Author Response · Authors · 2022-11-15
> > > **GSS with 12 layers**
> > >
> > > We are delighted that we were able to address most of your concerns!
> > >
> > > We thank the reviewer for increasing the score (although just to make sure, we still see the old score of 5 on our end, it would be great if you could take another look to update it)
> > >
> > > ```depth vs width```.  Based on your feedback on the last remaining concern, we also ran an additional experiment where we compare DSS, GSS and GSS-12L (with increased MLP size). These results are in Appendix Section D of the new revision.
> > >
> > > For GSS-12L, as suggested, we reduced the number of layers from 16 to 12 and increased the MLP size from 4096 to 6144 in order roughly match the parameter count. We found that GSS-12L is faster yet almost as competitive as GSS (16 layers) on all 3 datasets.
> > >
> > > Definitely let us know if you had any other concerns. Also, if we have sufficiently resolved all of your concerns, we would really appreciate it if you are able increase your score to "accept" for strongly supporting our work!

---

> > > > ### Comment · Reviewer_Rso2 · 2022-11-15
> > > > **Response**
> > > >
> > > > Thanks for your quick response. This additional experiments addressed my last concern.
> > > >
> > > > There was an issue of the OpenReview system, and I cannot update my score. Now I have upgraded my score to 6.

---

> > > > > ### Author Response · Authors · 2022-11-18
> > > > > **Thank you!**
> > > > >
> > > > > Thank you for your prompt response and updating the score! Definitely let us know if in case you had any other lingering concerns. We would be more than happy to address them.

---

### Official Review · Reviewer_GxKh · 2022-10-26

**Confidence:** 4
**Correctness:** 3
**Technical Novelty And Significance:** 2
**Empirical Novelty And Significance:** 2
**Recommendation:** 5

**Clarity, Quality, Novelty And Reproducibility:**

This paper is clear and provides details on hyperparameter settings. This work is original to my knowledge.

**Strength And Weaknesses:**

Strengths:
1. Shows that a simple change that applies S4 to a space of reduced dimensionality improves throughput by a factor of two to three.
2. It's interesting that a model trained on shorter context sizes can learn to utilize longer contexts at inference time.

Weaknesses:
1. The main argument is that while the proposed model underperforms the baseline block-recurrent transformer at similar number of parameters, it's better at the same training cost. However, it is not clear to me that the baseline is optimized for limited training cost scenarios: for example, does training the baseline for a shorter period of time significantly degrades its performance in terms of PPL? In this regard, I think showing a curve that plots validation PPL against training time might be more convincing. But even so, the comparison is still not fair if the baseline is not tuned to arrive at the Pareto frontier for the given compute (such as by changing model size, learning rate, and other hyperparameters)
2. The model architecture seems to be optimized for a particular hardware architecture TPUs (which is not that widely accessible compared to GPUs). Does the increased throughput generalize to different hardware architectures such as GPUs?
3. Recent works on training large LMs (such as OPT) show that the attention operation only takes negligible time for large models, and the dominating term is the feedforward operations. Is it still meaningful to address the quadratic attention complexity issue as hardware gets more and more powerful and we can run larger and larger models?
4. Since Gated Attention Units (Hua et al 2022) motivated this work, why don't you compare to it as a baseline? Besides, in Table 1, can you compare to Transformers on other datasets as well (for the 512 setting) to get a sense of how much gain we can get by leveraging more context?

Typos:
1. Page 6 last paragraph: we present results for on a large range of sequence lengths

**Summary Of The Paper:**

The attention mechanism in transformers scales quadratically with the input size that prohibits its application to long sequences. This work proposes a variant of S4 that can scale to long sequences. The key change is to apply S4 to a space of reduced dimensionality before projecting it back to the original space inspired by a recent work Gated Attention Units. This change reduces the time of fast Fourier transform operations which is a bottleneck on TPUs, resulting in a 3X speedup. In addition, this work also found that random initialization works as well as Hippo initialization.  Experiments on long-form text generation datasets found that while the proposed model slightly underperforms baselines such as block-recurrent transformer at the same parameter count, the proposed model is better at a fixed training cost, especially when interleaved with transformer layers. Besides, the model trained on shorter context window sizes is able to generalize to utilize more context at test time.

**Summary Of The Review:**

I think this paper is marginally below the acceptance threshold, due to a few concerns: first, at the same parameter count this work seems to underperform baselines; second, I have some concerns about the argument based on limiting the training cost as mentioned above; third, the final model is a hybrid between S4 and transformer layers, and I'm not sure if it will be actually used due to 1) the hybrid architecture is not as easy to implement and tune as pure S4 layers or pure transformer layers; 2) the speedups are only measured on TPUs which is less widely used than GPUs; 3) the recent lesson we learned in training large LMs that attention cost is negligible as we scale the model size. That being said, this paper has its merits in that it fixes a compute bottleneck and observes good empirical speedups. Therefore, I think this paper is marginally below the acceptance threshold.

---

> ### Author Response · Authors · 2022-11-13
> **Response to Reviewer GxKh**
>
> Thank you for your comments and careful consideration! Below we discuss how we have addressed all of your concerns. We hope that our response alleviates some of your reservations and are able to support our work.
>
> 1. ```baseline is optimized for limited training cost scenarios```: This is a great point! Since the original Block-Recurrent Transformer (BRT) paper does not do a compute-optimal analysis (since this is rarely done), we are unable to conduct that comparison. However, GSS has several conceptual properties that make it advantageous compared to BRT, especially as the sequence length is increased. E.g. GSS could contextualize very long sequences due to its convolutional view and still decode at fast rates at inference time due to its recurrent view. BRT is inherently sequential and the performance in both tasks gets poorer with larger sequence length. Thus, our aim in Table 2 is just to illustrate that GSS can be quite effective even at training time at the sequence lengths that baselines work well on as reported by them. In fact, we were quite surprised that GSS can be this competitive even at training time.  To further support strong contextualization offered by GSS, we report very competitive results in comparisons to other baselines in Table 3 for PG-19 in an open-vocabulary setting.
>
> 2. ```attention operation only takes negligible time for large models``` : OPT was trained with input length of 2048 which is significantly shorter compared to the range of sequence lengths with which we evaluate in this work. Table 2 shows that indeed scaling to larger input lengths leads to lower perplexity. None of the large LMs have delivered encouraging performance on LRA and using more/faster hardware to sustain a $\Omega(L^2)$ blowup in compute would be prohibitive at best. For instance, we don't foresee vanilla Transformer would be performant on a sequence length of 65k or even 1M whereas GSS would be a natural fit in those constraints. Our hope is that GSS inspires future LLMs to operate on **extremely long** contexts.
>
> 3. ```the hybrid architecture is not as easy to implement```: do you see any specific difficulty in the implementation? We would like to argue that our formulation of hybrid GSS-Transformer is in fact one of the simplest ways of creating a hybrid between the two. We simply interleave vanilla Transformer blocks with GSS. There could be other richer ways to create a hybrid, which we believe is an interesting direction for future work.
>
> 4. ```GAU/FLASH as a baseline``` : This is a fair point and one which other reviewers (and even we) share! We implemented FLASH baseline in our setup and compared it with GSS on all 3 long range datasets we considered. Results are presented in Appendix Section B, Table 4. We strived hard to make the comparison fair and found that GSS (and GSS-Hybrid) is very competitive with FLASH, where the **GSS-Hybrid model outperforms FLASH on PG-19 and Github**. Since we used vanilla Transformer blocks in GSS-Hybrid, we believe replacing the Transformer blocks with GAU can further improve performance but leave this for future work.
>
> 5. ```Table 1, compare to Transformers on other datasets``` : This is a great suggestion! We **have updated Table 1 with results for Transformer using 512 context length** and can be found in the attached revision. As expected, on PG19, ArXiv and GitHub, GSS performs better as it is capable of leveraging a larger context but the extent to which it is able to do that seems to vary between datasets.
>
> 6. ```Typos: Page 6``` : fixed.
>
> 7. ```Benchmarking on GPUs```:  We **benchmarked GSS vs DSS on GPUs** but did not find as dramatic speedups as we did on TPUs as FFTs on GPUs are highly optimized whereas, currently, FFTs are slow to perform on TPUs. TPUs have been responsible for some of the most important models (T5, PaLM, etc) and it is *beneficial to design models optimized for TPUs as well*.
> We did the following benchmarking for models used in Table 1 on a 24GB Nvidia 3090 GPU using PyTorch. DSS has model dimension 1024, N=512 and has 12 blocks where each block contains 1) a DSS layer with GELU non-linearities followed by GLU with expansion 2, another layer identical to part 1) and, finally 3) a FF layer with intermediate dimension 4096. GSS stack has 16 identical GSS blocks as described in the paper for the experiments in Table 1. Vocabulary size was 32k. In the following table, B is batch size, L is input sequence length and we used gradient checkpointing (GC) to reduce memory footprint.
> |  | params | number of blocks | B=8, L=4096 w/ GC | B=4, L=8192 w/ GC | B=2, L=16384 w/ GC |
> |---|---|---|---|---|---|
> | DSS | 210M | 12 | 3.00 sec / step | 3.64 sec / step | 4.73 sec / step |
> | GSS | 193M | 16 | 2.40 sec / step | 2.51 sec / step | 2.68 sec / step |
> | speedup |  |  | 1.25x | 1.45x | 1.76x |

---

> > ### Author Response · Authors · 2022-11-18
> > **Gentle reminder**
> >
> > Hello Reviewer GxKh,
> >
> > As the discussion period is closing shortly, please let us know if you have any further questions or concerns!

---

### Official Review · Reviewer_Z4KM · 2022-10-27

**Confidence:** 4
**Correctness:** 4
**Technical Novelty And Significance:** 3
**Empirical Novelty And Significance:** Not applicable
**Recommendation:** 6

**Clarity, Quality, Novelty And Reproducibility:**

Paper is clearly written and reproducible based on the presented experimental details.

**Strength And Weaknesses:**

**Strength:**
1. The paper is very clearly written and easy to understand.
2. Authors find that the main bottleneck of DSS comes from the high contextualized input dimension, as the FFT is performed independently on each dimensions of the sequence. With the GAU gating design, GSS enables a 4x reduction in this dimension and increases the throughput when trained on TPUs.
3. GSS can extrapolate naturally to unseen sequence length at test time due to the recurrent nature of DSS layer, which also allows faster inference speed.


**Weaknesses**:
1. GSS is largely motivated by GAU, but GAU is not included as a baseline method in this paper. The hybrid model with interleaved Transformer layer is closely related to the SRU++ (When Attention Meets Fast Recurrence: Training Language Models with Reduced Compute) paper, but it is not considered as a baseline method either.

**Questions**:
1. GSS is only evaluated on the long range language modeling tasks, which is fair enough. But have you evaluated GSS on any other tasks or commonly used LM datasets, e.g. Wikitext-103 and LRA?
2. Have you measured the training speed on GPUs?

**Summary Of The Paper:**

This paper proposes a new layer named Gated State Space (GSS) that combines the gate design in Gated Attention Unit (GAU) with a simplified version of Diagonal State Space (DSS) layer. Authors further propose a hybrid model that combines GSS with sparingly interleaved Transformer blocks. The resulted model trains significantly faster than the original DSS and can extrapolate to longer sequences at test time. The proposed model is evaluated on 4 long range language modeling tasks, comparing with a series of strong baseline models.

**Summary Of The Review:**

This paper is an established work that is based on some previous works, such as GAU and DSS, speeding up training compared to DSS. However, the empirical improvements seem not be very strong compared to the baseline methods. But it's a good paper that contributes to the state space models and connects with Transformer layers.

---

> ### Author Response · Authors · 2022-11-13
> **Response to Reviewer Z4KM**
>
> We thank the reviewer for the detailed review and insightful comments. We hope that our response alleviates your concerns and you would consider strongly supporting acceptance of our paper!
>
> Our response to the raised concerns are as follows:
> 1. ```GAU/FLASH as baseline``` : This is a fair point and one which other reviewers (and even we) share! To that end, we implemented FLASH baseline in our setup and compared it with GSS on all 3 long range datasets we considered. The results are presented in Appendix Section B, Table 4. We strived hard to make the comparison fair and found that similar to the comparison with Block Recurrent Transformer, GSS (and GSS-Hybrid) is very competitive with FLASH, where the **GSS-Hybrid model outperforms FLASH on PG-19 and Github**. Additionally, since we used vanilla Transformer blocks in GSS-Hybrid, we believe replacing the Transformer blocks with GAU can further improve performance, but due to resource constraints, we leave this for future work.
>
> 2. ```SRU++``` : We were simply not aware of this work! While we could not include this as a baseline, we have included a citation. We note that even though the element-wise/simple recurrence used in SRU/SRU++ is parallel in the hidden size dimension, it is *sequential in the time dimension during training* due to the use of non-linearities whereas DSS convolutions are parallel during training.
>
> 3. ```Wikitext-103/LRA``` : One of the main contributions of our work is in fact showing that SSMs such as DSS are effective on large-scale language modeling (LM). While DSS was shown to be effective on **LRA which consists of only sequence classification tasks**, it was unclear if it could provide speedups on LM and this is why we focused on autoregressive LM. In short, sequence-classification is not the focus of this work as the performance of DSS (i.e. the contextualizing component of GSS) on such tasks is well-established on LRA, speech recognition, etc.
>
> 4. ```Benchmarking on GPUs```:  We **benchmarked GSS vs DSS on GPUs** but did not find as dramatic speedups as we did on TPUs as FFTs on GPUs are highly optimized whereas, currently, FFTs are slow to perform on TPUs. Having said that, TPUs have been responsible for some of the largest and most important models such as T5, PaLM, etc and we believe it is *beneficial to design models optimized for TPUs as well*.
> We did the following benchmarking for the models used in Table 1 on a 24GB Nvidia 3090 GPU using PyTorch. DSS has model dimension 1024, N=512 and has 12 blocks where each block contains 1) a DSS layer with GELU non-linearities followed by GLU with expansion 2, another layer identical to part 1 and, finally 3) a FF layer with intermediate dimension 4096. GSS stack has 16 identical GSS blocks as described in the paper for the experiments in Table 1. Vocabulary size was 32k. In the following table, B is batch size, L is input sequence length and we used gradient checkpointing (GC) to reduce memory footprint.
> |  | params | number of blocks | B=8, L=4096 w/ GC | B=4, L=8192 w/ GC | B=2, L=16384 w/ GC |
> |---|---|---|---|---|---|
> | DSS | 210M | 12 | 3.00 sec / step | 3.64 sec / step | 4.73 sec / step |
> | GSS | 193M | 16 | 2.40 sec / step | 2.51 sec / step | 2.68 sec / step |
> | speedup |  |  | 1.25x | 1.45x | 1.76x |

---

> > ### Author Response · Authors · 2022-11-18
> > **Gentle reminder**
> >
> > Hello Reviewer Z4KM,
> >
> > As the discussion period is closing shortly, please let us know if you have any further questions or concerns!

---

### Official Review · Reviewer_LYuN · 2022-10-27

**Confidence:** 3
**Correctness:** 3
**Technical Novelty And Significance:** 3
**Empirical Novelty And Significance:** 3
**Recommendation:** 6

**Clarity, Quality, Novelty And Reproducibility:**

The experimental part lacks of analysis on the proposed Gated State Spaces to show the insight. Without showing that, we are not convinced why Gated State Spaces helps by some intuitive proves.

**Strength And Weaknesses:**

Strength:
1. The motivation of modeling long range dependencies is clear.
2. The paper is in a good shape and easy following.

Weaknesses:
1. The experimental part lacks of analysis on the proposed Gated State Spaces to show the insight. This part is not convincing.
2. Some minor issue in paper. For instance, I found it is hard to find the alignment on figure and code in Figure 1.


**Summary Of The Paper:**

This paper introduces GSS, a general purpose sequence model which leverages gated units and trains significantly faster as shown on several language modeling benchmarks. This simple-to-implement alternative to S4 and DSS which trains 2-3 times faster, and is competitive with Transformer-based baselines on long-range language modeling benchmarks.

**Summary Of The Review:**

The paper is well written and easy following but the experimental part lacks analysis. I think a minor revise is needed.

---

> ### Author Response · Authors · 2022-11-13
> **Response to Reviewer LYuN**
>
> We thank the reviewer for the review and encouraging comments. To address the main concern, we have **performed additional ablation experiments and added an analysis section** in the attached revision. This exercise has already led to interesting and counter-intuitive insights, thus we are quite thankful to the reviewer for the suggestion. We would really appreciate if you would consider increasing your score to “accept” if you find the analysis satisfactory, also definitely let us know if you have any other suggestions.
>
> Our response to the specific concerns are as follows:
> 1. ```Analysis of GSS layers``` :  That is a great suggestion! We agree that qualitative analysis of GSS would improve the quality of the paper. In the analysis section (Appendix Section C) we focus on 2 important hyperparameters:
>     - In the first set of our experiments we ablate the **kernel length** used in each GSS block. Intuitively, kernel length controls the window of context that a GSS layer is allowed to perform contextualization in.
>     - Since the hybrid between GSS and Transformer is our best performing model, to understand it better, we also perform the same experiment with GSS-Hybrid and find that hybrid version leads to less degradation in performance as kernel length is made smaller. Contrary to what we thought, this suggests that intermediate Transformer layers may be helping with modeling context lengths beyond the chunk length of 512 that it is limited to.
>     - Additionally, in GSS-Hybrid model, we ablate **attention chunk length**, where we find that even a very small chunk length of 64 leads to better performance than vanilla GSS model, suggesting that even a modest amount of attention can provide complementary benefit to GSS.
>
> 2. ```Alignment of Figure 1``` : We double checked the figure and found the pseudocode to be correct - could you please elaborate which line in the code was hard to associate with the Figure? We omit the residual connection / layer norm in the figure.

---

> > ### Author Response · Authors · 2022-11-18
> > **Gentle reminder**
> >
> > Hello Reviewer LYuN,
> >
> > As the discussion period is closing shortly, please let us know if you have any further questions or concerns!

---

### Decision · Program_Chairs · 2023-01-20

**Decision:**

Accept: poster

**Justification For Why Not Higher Score:**

The paper's empirical results are not comprehensive enough to justify a spotlight or oral presentation.

**Justification For Why Not Lower Score:**

The proposed method and the experimental results presented are strong enough to justify acceptance, even though the paper could benefit from some additional improvements.

**Metareview: Summary, Strengths And Weaknesses:**

The paper proposes the Gated State Space (GSS) layer for autoregressive sequence modelling. The approach is an alternative to Diagonal State Spaces (DSS), training much faster, in particular on TPUs, while scaling to longer sequence lengths and generalizing better on longer inputs. On perplexity, performance is worse than Transformers evaluating on sequence length 512, but better when on longer sequence lengths (which the Transformer cannot handle), and comparable to or better than FLASH (which incorporated gating in Transformers) on lengths 1024. A GSS-Transformer hybrid performs best on some evaluations. While there is a small performance increase on GPUs, the main performance increase is specific to TPUs, which somewhat limits the potential impact of the method, although hardware specific optimizations are still valuable. The Transformer baseline is not necessarily tuned to be at the Pareto frontier for the given compute, but the GSS still obtains strong performance and enables leveraging longer sequence lengths.

The contributions of the paper are strong enough to be accepted. However, the paper could benefit from improving the presentation and performing more extensive experiments, such as evaluating on downstream sequence or text generation applications.

Comments: Please integrate the additional results in the appendix (of the revised version) into the main paper. Please avoid using subjective language in the paper, e.g. “To our delight” (p2), “it is quite relieving” (p9).


**Note From Pc:**

if the above contains the word "oral" or "spotlight" please see: "oral" presentation means -> notable-top-5% and "spotlight" means -> notable-top-25%. As stated in our emails, we are disassociating presentation type from AC recommendations